# Canine osteosarcoma genome sequencing identifies recurrent mutations in *DMD* and the histone methyltransferase gene *SETD2*

Heather L. Gardner [1], Karthigayini Sivaprakasam[2], Natalia Briones[2], Victoria Zismann[2], Nieves Perdigones[2], Kevin Drenner[2], Salvatore Facista[2], Ryan Richholt[2], Winnie Liang[2], Jessica Aldrich[2], Jeffrey M. Trent[2], Peter G. Shields[3], Nicholas Robinson[4], Jeremy Johnson[5], Susan Lana[6], Peter Houghton[7], Joelle Fenger[8], Gwendolen Lorch [8], Katherine A. Janeway[9], Cheryl A. London[4,10] & William P.D. Hendricks [2,10]

Osteosarcoma (OS) is a rare, metastatic, human adolescent cancer that also occurs in pet dogs. To define the genomic underpinnings of canine OS, we performed multi-platform analysis of OS tumors from 59 dogs, including whole genome sequencing ($n = 24$) and whole exome sequencing (WES; $n = 13$) of primary tumors and matched normal tissue, WES ($n = 10$) of matched primary/metastatic/normal samples and RNA sequencing ($n = 54$) of primary tumors. We found that canine OS recapitulates features of human OS including low point mutation burden (median 1.98 per Mb) with a trend towards higher burden in metastases, high structural complexity, frequent *TP53* (71%), PI3K pathway (37%), and MAPK pathway mutations (17%), and low expression of immune-associated genes. We also identified novel features of canine OS including putatively inactivating somatic *SETD2* (42%) and *DMD* (50%) aberrations. These findings set the stage for understanding OS development in dogs and humans, and establish genomic contexts for future comparative analyses.

[1] Sackler School of Graduate Biomedical Sciences, Tufts University, Boston, MA 02111, USA. [2] Translational Genomics Research Institute, Phoenix, AZ 85004, USA. [3] College of Medicine, The Ohio State University, Columbus, OH 43210, USA. [4] Cummings School of Veterinary Medicine, Tufts University, Grafton, MA 01536, USA. [5] Broad Institute, Cambridge, MA 02142, USA. [6] Colorado State University, Fort Collins, CO 80525, USA. [7] University of Texas Health Science Center, San Antonio, TX 78229, USA. [8] Department of Veterinary Clinical Sciences, The Ohio State University, Columbus, OH 43210, USA. [9] Pediatric Oncology, Dana-Farber Cancer Institute, Boston, MA 02215, USA. [10] These authors contributed equally: Cheryl A. London, William P. D. Hendricks. Correspondence and requests for materials should be addressed to W.P.D.H. (email: whendricks@tgen.org)

Osteosarcoma (OS) is the most commonly diagnosed primary bone tumor in dogs and children. It is a biologically aggressive disease characterized by lytic and proliferative boney lesions and a propensity for lung metastasis. In people, OS is a relatively uncommon cancer, with 800–1000 cases diagnosed per year. This is in contrast to pet dogs, where the annual incidence exceeds 25,000 cases[1]. Although surgery and chemotherapy extend survival times, approximately 30% of pediatric OS patients die due to metastatic disease within 5 years and over 90% of canine OS patients succumb to metastatic disease within 2 years following diagnosis. Furthermore, despite advances in understanding the molecular and genetic underpinnings of human OS, patient outcomes have not improved for humans or dogs over the past three decades. As canine OS recapitulates many of the salient biologic and molecular features of human OS, particularly with respect to treatment-resistant metastatic disease, it affords a comparative model that can be used to interrogate novel therapies within a compressed timeline[1–3].

Human OS tumor genomes frequently bear complex somatic genomic rearrangements, localized hypermutation, and abundant copy number variations (CNVs), with relatively few point mutations[4–6]. For example, whole exome sequencing (WES) of 123 human OS tumors identified somatic mutations in TP53 and RB1 (47% combined) alongside candidate driver mutations in an additional 14 cancer genes including BRCA2, RET, ATM, PTEN, WRN, and ATRX. However, only TP53 and RB1 bore somatic mutation rates >3%[5]. Whole genome sequencing (WGS) studies have identified additional mutations not detectable by WES such as intronic translocations impacting TP53 and other cancer genes, supporting the notion that virtually all OS tumors bear inactivating TP53 mutations (95%) and that other driver mutations such as DLG2 (53%), ATRX (29%), RB1 (29%), and MDM2 (3%) are also more commonly altered[6]. While OS is genomically heterogeneous, many of the changes identified impact a few key signaling pathways, indicating that despite marked chromosomal instability and genomic heterogeneity, phenotypic convergence exists. For example, numerous preclinical and clinical studies in OS have demonstrated constitutive PI3K/mTOR signaling despite a relatively low prevalence of PIK3C and PTEN mutations[4,7].

Canine OS, although less well-studied, demonstrates numerous clinical and molecular similarities to human OS and has been leveraged as a spontaneous large animal disease model to help identify biomarkers and guide therapeutic development. Early cross-species clinical efforts involved the use of dogs with appendicular OS to optimize limb-sparing techniques and investigate novel treatment combinations to inform similar efforts in children[8]. More recently, a study comparing the transcriptional profiles of canine and human OS found them to be virtually indistinguishable. Notably, high IL-8 expression was observed in all canine OS samples leading to the subsequent finding that IL-8 overexpression is a poor prognostic indicator in human OS[9]. Genetic risk associated with OS in dogs has been particularly well-studied, facilitated by selective inbreeding and population bottlenecks within breeds that created long linkage disequilibrium. One genome-wide association study identified risk loci containing OS-associated genes, such as CDKN2A/B, AKT2, and BCL2. These loci explained 55–85% of the variance across the three breeds evaluated (Rottweilers, Irish Wolfhounds, Greyhounds)[10].

With respect to genomic drivers of canine OS, several similarities have been identified with human OS including mutations or copy number alterations in TP53 (24–75%), RB1 (29–61%), PTEN (42%), and MYC (40%), among others[4,7,10–21]. A comparative cross-species array comparative genomic hybridization (aCGH) study of canine and human OS identified copy number deletions in a novel tumor suppressor gene, DLG2, in 42% of human and

55.6% of canine OS samples[22]. Additionally, the canine and human tumors showed broad genomic similarity with recurrent copy number aberrations in oncogenes and tumor suppressor genes shared between both species (MYC, CDKN2A/B, RB1, PTEN)[22]. Recently, WES performed on matched primary tumor/ normal of canine OS within three predisposed pure breeds (Rottweilers, Golden Retrievers, Greyhounds) found TP53 to be most frequently mutated gene (83%), consistent with findings in human OS. Notably, the tumor suppressor SETD2, a histone methyltransferase, was also mutated in 21% of cases evaluated[14]. In people, SETD2 has predominantly been associated with a tumor suppressor function via inactivating mutation in clear cell renal cell carcinoma and hematologic malignancies, possibly through the effects of its loss on generation of genomic instability and unchecked transcriptional initiation[23]. While the biologic consequences of SETD2 mutations in canine OS are unknown, this work highlights the potential contribution of epigenetic modifications to OS pathogenesis.

Despite the previous body of work interrogating the canine OS tumor genome, several knowledge gaps remain. As many of the studies performed to date have involved only a few breeds, the somatic variations identified may be breed-specific and thus may not translate across the broader landscape of dogs that develop OS. Additionally, WGS of canine OS has not yet been performed, leaving its inherent structural complexity largely unexplored. Finally, there are few comparative studies of matched primary and metastatic lesions in either human or canine OS. Given the inherent genomic complexities underlying both human and canine OS, a better understanding regarding drivers of disease progression leading to metastasis would facilitate therapeutic development[16,21,24]. Therefore, we set out to perform comprehensive multiomic profiling of both primary and metastatic canine OS across multiple dog breeds with the goal of clarifying the genetic changes that orchestrate primary tumor growth as well as support the metastatic phenotype. Here we demonstrate that, concordant with human OS, canine OS is characterized by marked structural complexity and shared aberrations in key tumor suppressor genes and oncogenes including TP53, MYC and PI3K signaling pathways. Additionally, we demonstrate novel mutations in the histone methyltransferase SETD2 and DMD, the gene encoding dystrophin.

## Results

**Cohort characteristics.** We collected primary canine OS, matched normal samples, and matched metastases in three cohorts (Table 1): primary OS samples ($n = 24$) for WGS and RNA-seq; primary ($n = 13$) and metastatic ($n = 8$) OS samples for WES and RNA-seq; and primary OS samples for RNA-seq only (Fig. 1). Supplementary Data 1 list complete sequencing metrics and all tools are referenced in Supplementary Data 2.

Across the combined cohorts, patient demographics and clinical presentation were consistent with published data[25,26] (Supplementary Data 3). Median age at diagnosis was 7.7 years (range 1–12 years). A small subset of dogs developed OS at a young age as previously reported[27]. Metastatic disease developed primarily in the lungs (33/59) with a smaller percentage (10/59) occurring in other bones or subcutaneous or visceral tissues. Sixteen dogs did not have follow-up information on progression. It is therefore likely that the true incidence of metastatic disease is underrepresented in this study. Only three dogs presented with concurrent primary tumors and metastatic disease ($n = 3$ RNA-seq; $n = 2$ WES). The remaining samples were obtained at the time of limb amputation prior to initiation of chemotherapy. This reflects the typical clinical situation in which <10% of dogs with appendicular OS present with macroscopic metastases. Mixed

**Table 1 Sample cohort and sequencing platforms**

| Tumor source | WGS (n = 24) | WES (n = 13) | | RNA-seq (n = 54) |
| --- | --- | --- | --- | --- |
| | Primary | Primary | Metastatic | Primary |
| Germline variants | X | X | | |
| Somatic SNVs | X | X | X | |
| Mutational signatures | X | X | | |
| Somatic SVs | X | | | |
| Copy number variants | X | | | |
| Differential expression and unsupervised hierarchical clustering | | | | X |
| Pathway enrichments | | | | X |

*WGS* whole genome sequencing, *WES* whole exome sequencing, *SNV* single nucleotide variant, *SV* structural variant

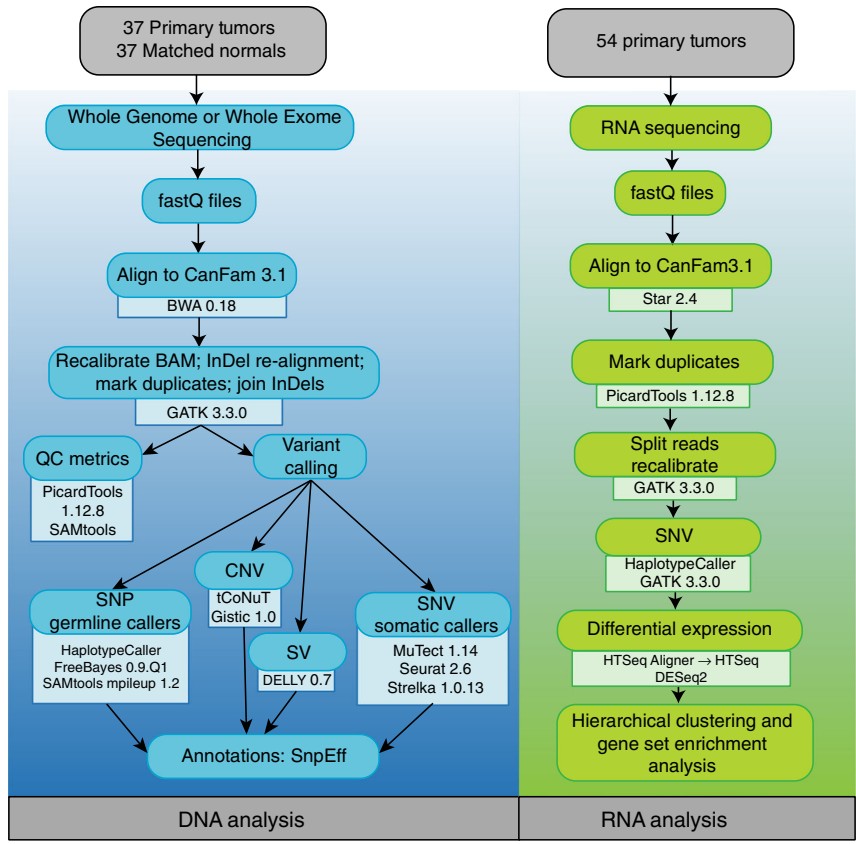

**Fig. 1** Analysis Pipeline for WGS, WES and RNA-seq. Flow-chart demonstrates sequential use of tools in evaluation of DNA and RNA samples

breed dogs (9/59), Greyhounds (9/59) and Golden Retrievers (8/59) were the most common breeds represented across all samples. However, a variety of giant and large breed dogs were included, providing substantial diversity within the samples analyzed. We did not observe any correlations between individual dog breeds and driver mutations described below, although given the breed heterogeneity in our cohort, this study is not powered to detect moderate- or low-frequency breed-specific events.

**Germline variants identified in canine OS**. We evaluated WGS and WES data from constitutional DNA for coding germline variants in 28 genes previously associated with both canine and human OS (Supplementary Data 4) as well as variants in *SETD2* and *DMD* (specific to canine OS) that are described in detail below[14,28]. Germline variants were identified in tumor suppressor genes including *APC2* (10/24 WGS; 1/13 WES), *BLM* (9/24 WGS), *BRCA2* (23/24 WGS), *TP53* (1/24 WGS), *RB1* (14/24

WGS; 6/13 WES), *WRN* (13/24 WGS; 6/13 WES) and *CDKN2B* (10/24 WGS; 5/13 WES), which include predicted damaging frameshift and missense variants (Supplementary Data 5). While *CDKN2B* mutations were not associated with age of diagnosis, downregulation of CDKN2A/B was identified in the transcriptome, suggesting that the germline disruptive inframe deletions are damaging alterations (Supplementary Data 5). Germline variants were also identified in *SETD2* (1/24 WGS) and *DMD* (2/24 WGS); however, the significance of these alterations is unknown. *DMD*, the gene encoding dystrophin, connects the cytoskeleton with the extracellular matrix and somatic loss has been associated with a tumor suppressor function in several human cancers[29,30].

**Somatic mutation burden and mutation signatures in canine OS**. Based on WGS, the median somatic mutation rate (mutations per callable sequenced haploid megabase) of all coding and

noncoding single nucleotide variants (SNVs) was 1.98. A median of 23 somatic coding mutations (25 WGS, range: 3–63; 11 WES, range: 0–129) was identified in the primary tumor samples (Supplementary Data 6). Two samples analyzed via WES demonstrated a higher mutation rate in comparison to the remaining samples. While these samples were obtained from older dogs, no other distinguishing explanatory clinical or genetic features were identified. The median allele frequency of all somatic SNVs called by at least two callers was 0.17 (range: 0.04–1) in WGS and 0.49 (range: 0.03–1) in WES samples. Tumor content assessment was similar between WGS and WES samples. Missense mutations were the most frequently represented somatic coding point mutation type in each tumor sample (Figs. 2, 3a, Supplementary Data 6). A subset of samples (4/24 WGS; two of which are shown in Fig. 3d) exhibited localized hypermutation characterized by C > T substitutions, consistent with kataegis (Fig. 3d; Supplemental Fig. 1). The trinucleotide context of somatic SNVs was also evaluated to assess mutation signatures and mutation etiology across all samples using a Bayesian non-negative matrix factorization method. Mutation signatures were first analyzed according to published signatures of mutation processes and listed herein based on the corresponding COSMIC terminology[31]. The most common base change identified in all samples was C > T within the CpG trinucleotide context, corresponding to the COSMIC 1 signature associated with aging, which is prevalent in most human cancers (including OS) and is the result of spontaneous deamination of 5-methylcytosine (Supplemental Figs. 2, 3)[31].

Distinct mutation signatures were identified when comparing the WGS and WES datasets. When the WGS samples were analyzed alone, two signatures were detected corresponding to COSMIC 1B and COSMIC 9 (Supplemental Fig. 2). The COSMIC 9 signature is believed to be secondary to polymerase η processing cytidine deamination and has been reported in hematologic malignancies in people[31]. The WES dataset included signatures corresponding to COSMIC 1A/B, COSMIC 17, COSMIC 5, and COSMIC U2 (Supplemental Fig. 3). The biologic significance of the U2 signature is ill-defined in human cancers, with a low probability of C > A, C > G, C > T, T > A, T > C and T > G substitutions. Signature 5 is characterized by transcriptional strand bias for T > C mutations. Finally, the COSMIC 17 signature is characterized by increased T > G mutations.

Assessment of somatic copy number variants (focal homozygous deletions or ≥2-copy gain somatic copy number variants using a Log2 fold change of ≤−0.9 and ≥0.4) from the 24 WGS tumor/normal pairs (Figs. 2, 3c, Supplementary Data 7) revealed focal copy number gains in CFA 13 involving PDGFRA (29% WGS) and MYC (38% WGS). Copy number losses were also identified in DLG2 (8% WGS samples), SETD2 (4% WGS), and DMD (29% WGS), genes previously associated with other human cancers such as OS, renal cell carcinoma and myogenic sarcomas[22,29,32]. Thirteen percent of samples did not show high-level focal copy number gains or losses based on the analysis and cutoffs described above. We did not observe a correlation between mutation burden or copy number aberrations and histologic tumor content, supporting that the absence of detection of high-level CNVs in these samples was not solely due to low tumor content (Supplementary Data 8). We also assessed more subtle copy number variants (Log2FC ≤ −0.05 and ≥0.5) in important OS genes including CDKN2A/B, DLG2, SETD2, and DMD. Using these guidelines, CDKN2A/CDKN2B copy number losses were identified in 46%, DLG2 loss in 37%, RB1 loss in 29%, PTEN loss in 45%, DMD loss in 50%, and SETD2 losses in 25% of WGS samples (Supplementary Data 9). GISTIC analysis of significant genomic regions recurrently impacted by somatic copy number changes based on amplitude, frequency, and

chromosomal boundaries of these events also confirmed the significant recurrence of several alterations identified with the tCoNuT algorithm, including copy number loss of CDKN2B and PTEN (Supplementary Data 10).

We identified at least one somatic translocation in 22/24 of primary OS tumors (Fig. 2b). A median of nine complex chromosomal translocations (range: 0–36), 10.5 deletions (range: 0–31), 9.5 duplications (range: 0–21) and 13.5 inversions (range: 0–40) were identified in this sample set (Fig. 3b). Supplementary Data 11 detail genes impacted by structural variants (SVs) across the WGS samples. Additionally, in 9/24 WGS samples we found chained rearrangements and complex chromosomal rearrangements involving multiple chromosomes that correlated with CNVs, suggestive of chromothripsis[33] (Supplemental Fig. 4).

WES analysis of somatic mutation burden in the ten matched primary/metastatic/normal samples demonstrated a trend towards higher somatic mutation burden in metastases, although the difference was not statistically significant. The primary tumors carried a median of 1.38 coding mutations/Mb and the metastatic tumors carried a median of 2.85 coding mutations/Mb ($p = 0.36$, Mann−Whitney U). Most primary tumors in this cohort (8/10) were collected in the absence of metastatic disease, with matched metastases collected at later timepoints. In the two matched primary/metastatic tumor pairs collected simultaneously in the setting of advanced metastatic disease (E3, E5), the mutation burden was higher in the corresponding metastatic lesion. These increases in mutation burden are unlikely to be due solely to treatment with DNA-damaging agents given that only one dog received one dose of a DNA-damaging agent (carboplatin chemotherapy) prior to collection of the metastatic lesion. We observed a spectrum of candidate pathogenic somatic point mutations in cancer genes in this set of samples (Supplemental Fig. 5, Supplementary Data 12) including shared and private mutations predominantly in TP53 (8/10) in addition to individual cases with ARID1B, DNMT1, KMT2D, POLG, PPM1D, PREX2, RB1, or RET mutations. In 3/10 matched samples, TP53 missense mutations were present in both the primary tumor and matched metastasis with an allele frequency (AF) of >0.6. A TP53 splice site variant was also shared at high AF in both a primary and matched metastasis in a fourth case. In one case, a TP53 mutation was present in the primary tumor, but not in the corresponding metastasis, while in three cases TP53 mutations were gained in the matched metastases. Acquired somatic point mutations in ARID1B, DNMT1, KMT2D, PPM1D, and RB1 were also noted in metastases, but not in their matched primary lesions lesions only. Overall, 4/10 metastases showed metastasis-specific acquisition of a likely pathogenic driver mutation relative to its matched primary lesion.

**Differential expression of immune response genes**. RNA-seq was performed in 54 primary OS tumor samples. Differential expression analysis was performed using DESeq2 for the 24 samples that also underwent WGS against a control canine osteoblast cell line. Normalized HTSeq counts for significantly differentially expressed genes were then used to perform hierarchical clustering across all 54 samples (Supplemental Fig. 6, Supplementary Data 13). When all genes were considered together, a cluster of 31 genes involved in aspects of immune response was shown to be underexpressed among most of the tumor samples (Supplementary Data 14 and 15) including those involved in chemokine and cytokine signaling (chemokine receptors-2 and -5, interleukin-31 receptor A, toll-like receptor 7), complement activation (complement C1q B-chain, complement C3a receptor 1, complement factor properdin) and caspase-mediated apoptosis (caspase-12, caspase recruitment domain

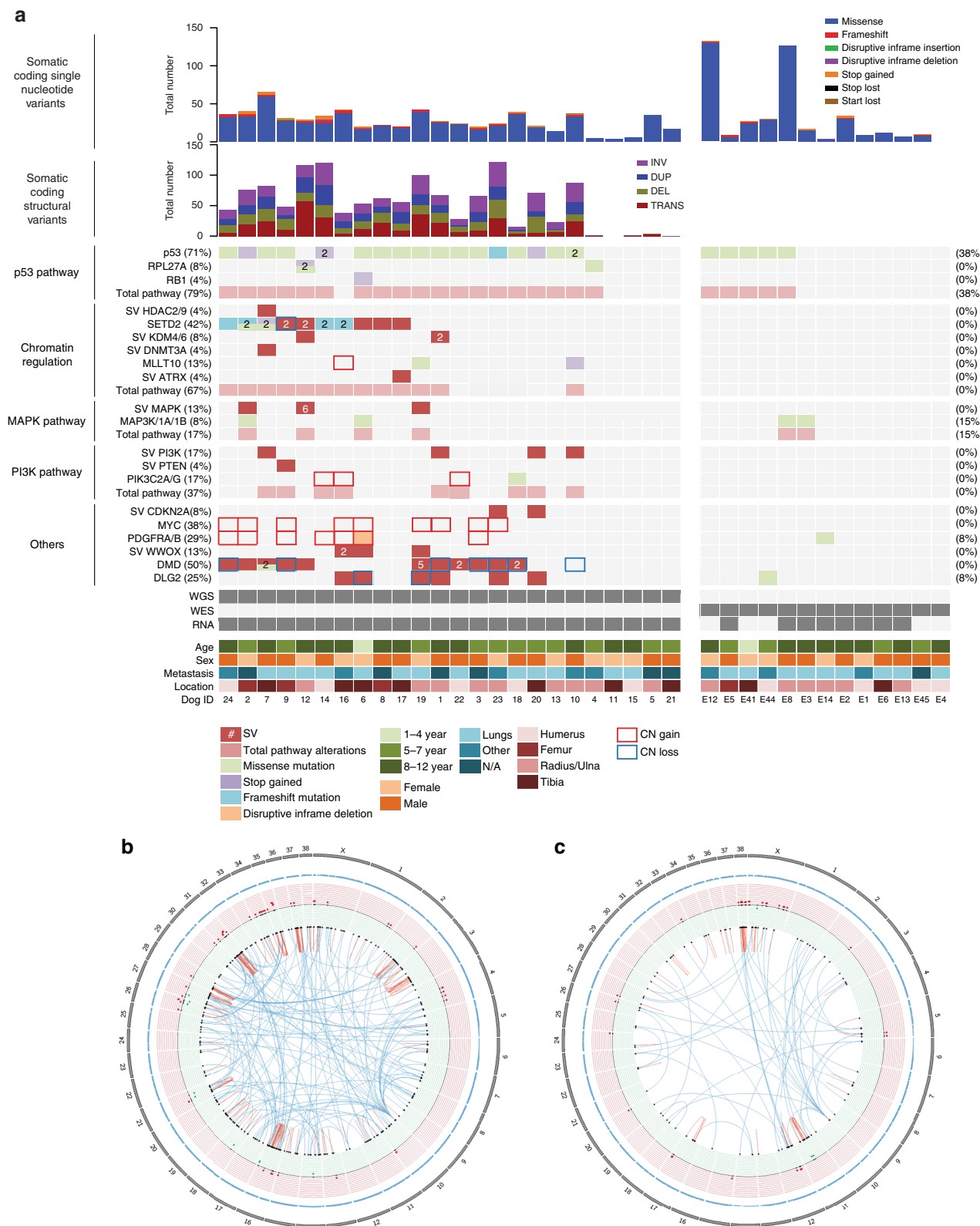

**Fig. 2** Recurrently mutated genes in canine osteosarcoma. **a** Single nucleotide variants were reported in samples subjected to both WGS and WES. Copy number variants (Log2FC < −0.9 and >0.4) and structural variants were reported in WGS samples. All mutations were clustered based on mutational burden in genes associated with chromatin/histones, *TP53* and *DMD*. **b**, **c** Circos plots on DogID #14 and #18. Blue triangles = SNVs; red dots = amplifications; green dots = deletions; dark red arrows = intra-chromosomal translocations; dark blue arrows = inter-chromosomal translocations. WGS whole genome sequencing, WES whole exome sequencing, SNV single nucleotide variant

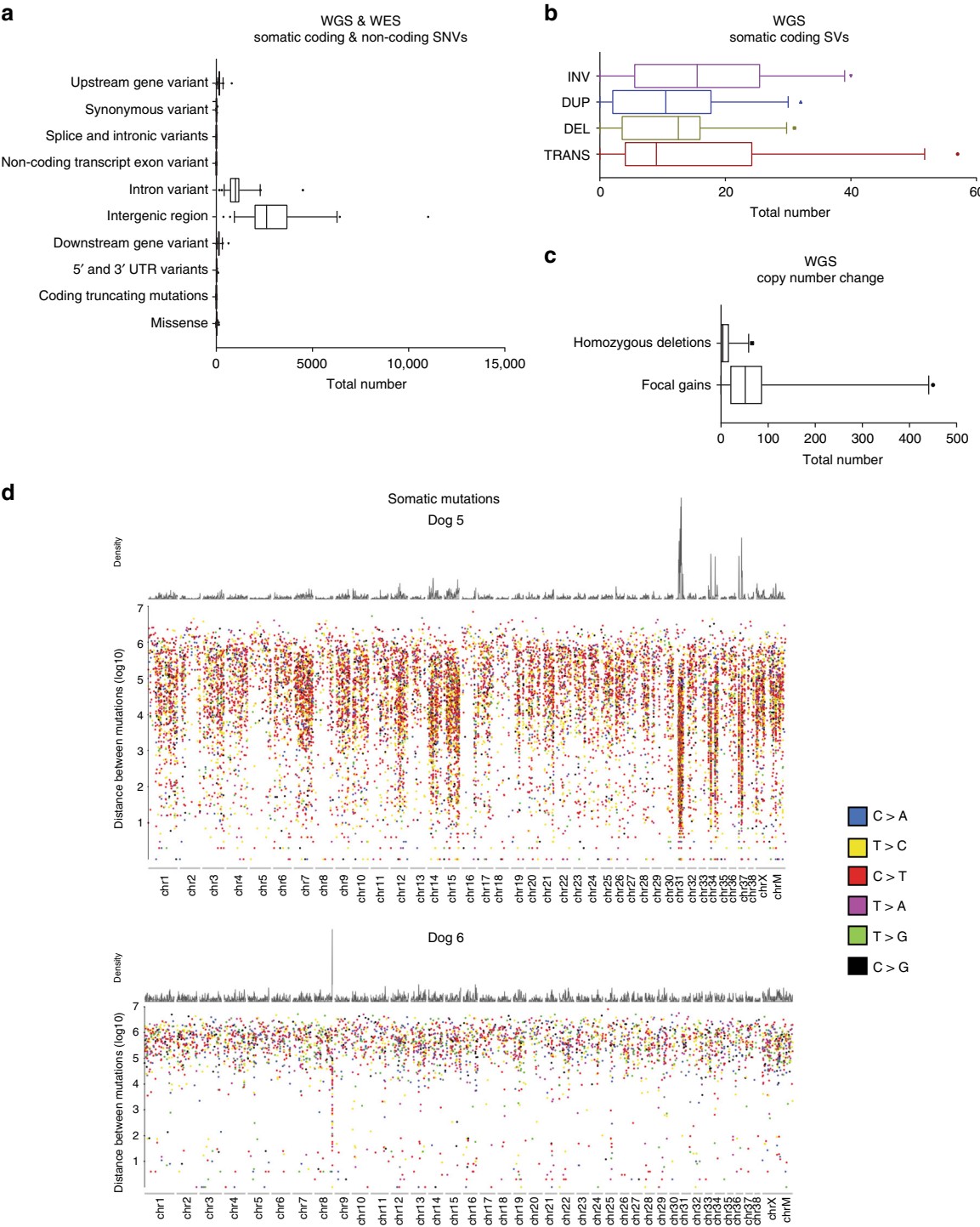

**Fig. 3** Somatic CNVs, SNVs, and SVs. Frequency of somatic coding and non-coding CNVs, SNVs and SVs across both WGS and WES samples. **a** WGS and WES somatic coding and non-coding SNVs. **b** WGS somatic coding SVs. Whiskers represent the 5-95th percentile. **c** WGS copy number changes. Whiskers represent the 5-95th percentile. **d** Rainfall plots illustrating the density and distribution of somatic mutations in two WGS samples. Base-pair distance between events represented on the y-axis

family member-11). Notably, programmed cell death 1 ligand 2 (PDL2) expression was low in 61% (33/54) of these samples. To further interrogate how these genes segregate into the two major sample clades based on hierarchical clustering, we compared the two major clades (clade 1 and clade 2) using DESeq2 analysis. Pathways involved in cytokine−cytokine receptor interaction, natural killer cell-mediated cytotoxicity, and T-cell receptor signaling (including PDL2) were found to be among the most significantly differentially expressed (upregulated in clade 1 and downregulated in clade 2) (Supplementary Data 14). However, differential gene expression in this analysis did not correlate with any clinical or genetic features of this cohort including the presence of mutations, structural variants or copy number aberrations in key genes mutated in this cohort (*RB1, SETD2, MLLT10, PTEN, PIK3C2A, PIK3C2G, CDKN2A, MYC, PDGFRA, PDGFRB, DMD,* and *DLG2*). Additionally, based on the assessment of correlation between gain-of-function or loss-of-function mutations in cancer genes bearing recurrent somatic structural

mutations in our cohort (CNVs and SVs) and gene expression values (TPMs), we identified significant correlation of low *SETD2* expression with somatic *SETD2* mutations ($p = 0.0003$) as well as significant correlation of increased *MYC* expression with somatic *MYC* copy number gains ($p = 0.018$). Other events not measured here may also affect dysregulation of some of these important cancer genes (e.g. epigenetic marks) and drive over- or under-expression such as that seen in a number of cases with relatively low levels of CDKN2A, but no identified inactivating mutation.

**Recurrently mutated genes in canine OS.** Somatic point mutations were most commonly identified in *TP53* (71% WGS, 38% WES), with point mutations and SVs in *SETD2* (42%) and *DMD* (38%) also occurring frequently in the WGS dataset (Fig. 2a). Missense mutations were the most prevalent somatic point mutation found in *TP53*, with frameshift (4%) and stop gained (13%) mutations noted less commonly. Aberrations in other known oncogenes and tumor suppressor genes were detected in MAPK and PI3K pathways, as well as in *MYC*, *PDGFRA/B* and *DLG2*. Specifically, point mutations, SVs and copy number aberrations in the MAPK and PI3K/mTOR pathways were found in 17% and 37% of the WGS samples, respectively (Fig. 2a). Aberrations in the PI3K/mTOR pathway involved *PIK3C2G* point mutations (4%), *PTEN* deletions (4%) and *PIK3CB/PIK3C2G* gene fusions (8%). Notably, we also identified several novel mutations in the histone methyltransferase *SETD2* and in *DMD*[34].

**TP53 is recurrently mutated in canine OS.** The p53 protein shares 79.8% amino acid identity between canine and human. Point mutations (missense, stop gain, frameshift) were the primary mutation type impacting *TP53* in both primary and metastatic OS samples: 17/24 of WGS cases, 5/13 of primary WES samples, and 6/10 of metastatic WES samples. Many of the missense mutations we observed in this cohort correspond to human-equivalent hotspots that are known to be pathogenic, such as codons 273, 282 and 285 in human *TP53* (Supplemental Table 6). No SVs or homozygous CN losses were identified in *TP53*. This is consistent with previous reports of *TP53* alterations in canine OS, in which point mutations predominate, although prior assessment of translocations in canine OS has been limited[14]. In contrast, both point mutations and structural variants in *TP53* are frequently noted in human OS[4].

**Canine OS exhibits recurrent mutations in SETD2.** Somatic point mutations, deletions, and chromosomal translocations were identified in *SETD2* (42%) in the samples that underwent WGS (Figs. 2a, 4, Table 2). Notably, there was no overlap between samples with point mutations or structural variants in *SETD2*. Additionally, one missense germline *SETD2* mutation was identified in a dog without somatic *SETD2* aberrations, but without a clear concomitant somatic mutation. The SETD2 protein shares 91.8% amino acid identity between canine and human. One frameshift mutation and one stop gained mutation correspond to human-equivalent regions with increased mutations (codons 1666 and 2077) in SETD2 (Supplementary Data 6).

**DMD is recurrently mutated in canine OS.** There is 94.1% homology between the canine and human DMD protein. Somatic *DMD* aberrations were noted in 50% of tumor samples that underwent WGS. DMD mutations were predominantly copy number losses and translocations. Somatic missense mutations and SVs were noted across 4% and 54% of WGS samples, respectively (Figs. 2a, 5, Table 3). The one missense mutation identified was not located in a known human-equivalent

mutation hotspot (Supplementary Data 6). An additional two germline *DMD* SNVs (8%) were detected in the WGS samples. SVs were composed of deletions ($n = 10$), inversions ($n = 1$) and chromosomal rearrangements ($n = 6$). All SV start and end sites were located in intronic regions of the gene and were within the first 63 exons of *DMD*. Copy number loss surrounding the *DMD* locus was also identified in five cases. While mutations in *DMD* have typically been associated with muscular dystrophy, loss of dystrophin has recently been associated with aggressive behavior in human cancers with and without myogenic programs, suggesting it may have tumor suppressor functions[29,35].

**Discussion**

The increasing use of targeted therapeutics and immune modulators in cancer treatment necessitates an improved understanding regarding the molecular and genomic nuances of individual tumors to most effectively advance clinical outcomes. This is particularly important for cancers such as human OS, a genomically heterogeneous disease with an array of complex molecular aberrations. As canine OS is often used as a spontaneous large animal model of human OS to interrogate novel therapies, developing a comprehensive understanding regarding its genomic landscape is critical. However, the genetic landscape of canine OS has not undergone detailed interrogation across a broad range of dog breeds[36]. Mutations in *TP53* and *RB1* as well as multiple members of the PI3K/AKT and MAPK signaling pathways were recently highlighted in human OS[4]. In canine OS, these same genes and pathways have been shown to be commonly altered, while somatic mutations in the histone methyltransferase *SETD2*, and germline variants in the cyclin-dependent kinase inhibitor *CDKN2A/B* were recently identified in tumor samples from Rottweilers, Greyhounds and Golden Retrievers[14]. Here, we sought to define the genomic landscape of canine OS using a combination of WGS, WES and RNA-seq to provide a more comprehensive characterization of actionable genomic aberrations, generating a body of data for use in comparing genomic drivers between dogs and humans.

Our multiomics approach demonstrates many similarities to published data regarding the human OS tumor genome and to the recent WES analysis undertaken in canine OS[4,14]. Consistent with human OS, the mutation burden in canine OS is relatively low in the context of all human cancers. However, with respect to pediatric tumors, the mutation burden is considered relatively high[37]. CNVs were also comparable across species. Gains in CFA13 involving the *MYC* and *PDGFR* loci were the most frequently documented gains in our dataset, concordant with prior reports[16,38]. Similarly, alterations in MYC have been associated with disease biology through effects on the MAPK pathway in human OS[39,40]. *TP53* missense mutations were the most prevalent SNV, while deletions and chromosomal translocations most commonly involved *SETD2* and *DMD*. While dogs largely develop OS in adulthood, the similar genomic features and clinical disease characteristics underscore the notion that age does not distinguish canine OS from the disease in children.

As in human OS, mutations tend to converge on pathways, and do not necessarily converge at one genomic locus. Pathways commonly altered in human OS were similarly affected in our canine OS dataset, including those involving TP53/RB, PI3K/AKT and MAPK[4]. As expected, point mutations were the most frequent alterations observed in *TP53* although the prevalence was higher in our analysis compared to previous publications (24–59%)[11–14,41,42]. While *TP53* gene alterations in canine OS predominantly consist of point mutations, in human OS both SVs and SNVs are typically found[4]. Notably, evaluation of chromosomal translocations in canine OS has been limited and we

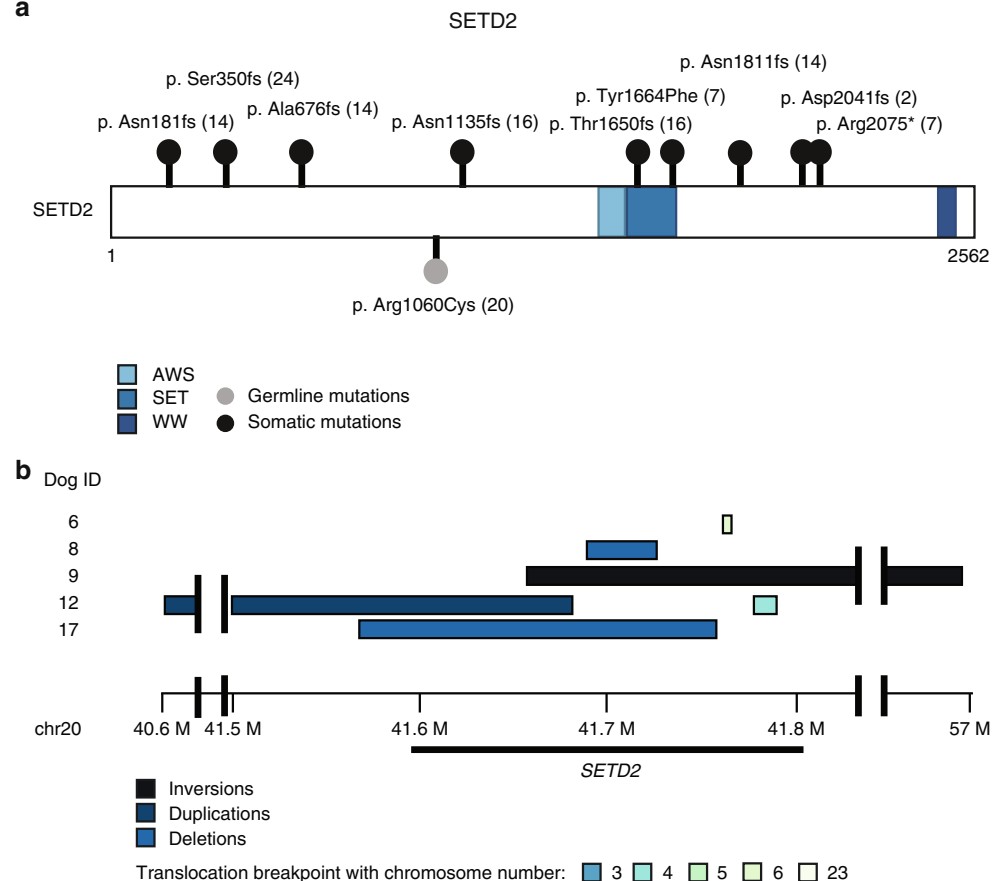

**Fig. 4** *SETD2* mutation burden in primary canine OS. **a** Lollipop plot demonstrating location of *SETD2* single nucleotide variants in canine OS. **b** Recurrent Copy Number Variations and Structural variants mutations found in five tumors defined by the region surrounding *SETD2*. **c** All somatic and germline SNV and SV calls affecting *SETD2*

### Table 2 SNV and SV calls affecting *SETD2*

**SETD2 single nucleotide variants**

| DOG ID | Chromosome | Position | Reference allele | Alternate allele | Effect | Status | Coding nucleotide | Amino acid |
|---|---|---|---|---|---|---|---|---|
| 2 | 20 | 41,762,908 | A | AG | Frameshift | Somatic | c.6121dupG | p.Asp2041fs |
| 7 | 20 | 41,723,954 | A | T | Missense | Somatic | c.4991A > T | p.Tyr1664Phe |
| 7 | 20 | 41,763,013 | C | T | Stop gained | Somatic | c.6223C > T | p.Arg2075* |
| 16 | 20 | 41,713,105 | CAG | C | Frameshift | Somatic | c.3399_3400delGA | p.Asn1135fs |
| 16 | 20 | 41,723,904 | G | GT | Frameshift | Somatic | c.4947dupT | p.Thr1650fs |
| 24 | 20 | 41,710,750 | C | CA | Frameshift | Somatic | c.1046dupA | p.Ser350fs |
| 14 | 20 | 41,711,731 | CT | C | Frameshift | Somatic | c.2022delT | p.Ala676fs |
| 14 | 20 | 41,737,948 | TA | T | Frameshift | Somatic | c.5432delA | p.Asn1811fs |
| 21 | 20 | 41,712,888 | C | T | Missense | Germline | c.3178C > T | p.Arg1060Cys |

**SETD2 structural variants**

| DOG ID | Chromosome start | Start position | Chromosome end | End position | SV type | Status | Effect |
|---|---|---|---|---|---|---|---|
| 6 | 20 | 41,777,850 | 6 | 63,389,989 | BND | Somatic | Transcript ablation |
| 8 | 20 | 41,707,519 | 20 | 41,725,779 | DEL | Somatic | Exon loss |
| 9 | 20 | 41,677,050 | 20 | 57,146,991 | INV | Somatic | Inversion |
| 12 | 20 | 40,619,935 | 20 | 41,682,808 | DUP | Somatic | Frameshift & splice variant |
| 9 | 20 | 41,793,070 | 20 | 44,196,596 | INV | Somatic | Bidirectional gene fusion |
| 12 | 20 | 41,797,955 | 4 | 51,687,909 | BND | Somatic | Gene fusion & frameshift |
| 17 | 20 | 41,586,296 | 20 | 41,760,821 | DEL | Somatic | Feature ablation |

*SNV* single nucleotide variant, *SV* structural variant

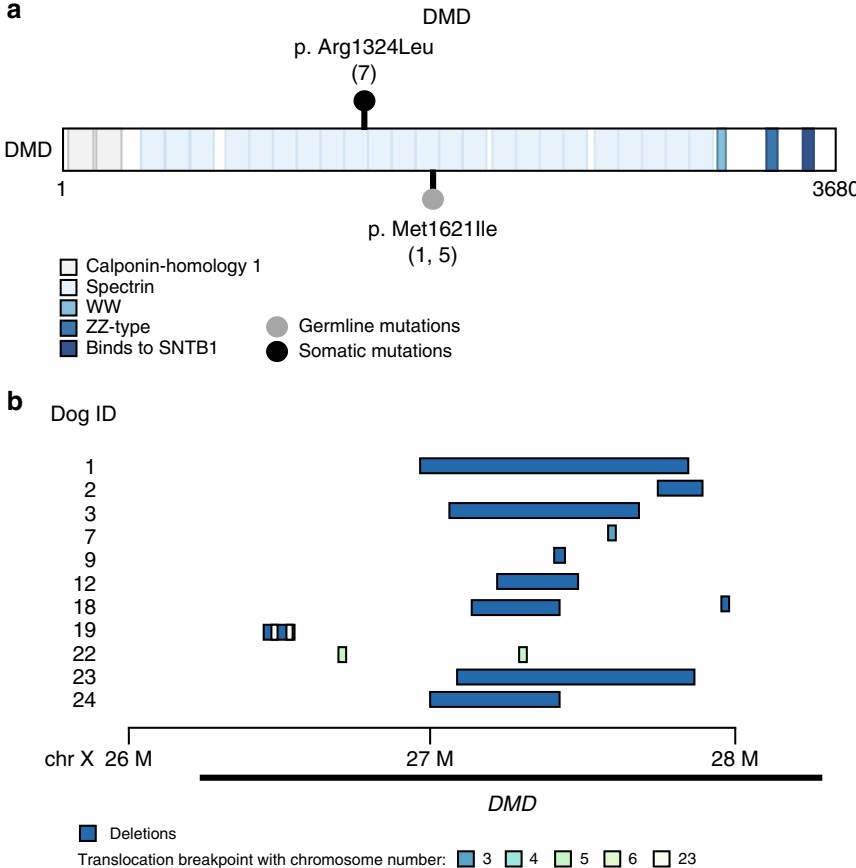

**Fig. 5** *DMD* mutation burden in primary canine OS. **a** Lollipop plot demonstrating location of *DMD* single nucleotide variants in canine OS. **b** Recurrent Copy Number Variations and Structural variants found in eleven tumors defined by the region surrounding DMD

## Table 3 SNV and SV calls affecting *DMD*

**DMD single nucleotide variants**

| DOG ID | Chromosome | Position | Reference allele | Alternate allele | Effect | Status | Nucleotide | Amino acid |
|---|---|---|---|---|---|---|---|---|
| 7 | X | 27,569,870 | C | A | Missense | Somatic | c.3971G>T | p.Arg1324Leu |
| 1 | X | 27,498,389 | C | T | Missense | Germline | c.4863G>A | p.Met1621Ile |
| 5 | X | 27,498,389 | C | T | Missense | Germline | c.4863G>A | p.Met1621Ile |

**DMD structural variants**

| DOG ID | Chromosome start | Start position | Chromosome end | End position | SV type | Status | Effect |
|---|---|---|---|---|---|---|---|
| 1 | X | 26,960,523 | X | 27,871,686 | DEL | Somatic | Exon loss |
| 2 | X | 27,734,857 | X | 27,925,969 | DEL | Somatic | Exon loss |
| 3 | X | 27,089,405 | X | 27,719,316 | DEL | Somatic | Exon loss |
| 9 | X | 27,460,863 | X | 27,470,244 | DEL | Somatic | Intron variant |
| 12 | X | 27,262,461 | X | 27,477,694 | DEL | Somatic | Exon loss |
| 18 | X | 27,198,484 | X | 27,466,712 | DEL | Somatic | Exon loss |
| 18 | X | 27,980,161 | X | 27,988,731 | DEL | Somatic | Intron variant |
| 19 | X | 26,528,219 | X | 26,529,125 | DEL | Somatic | Intron variant |
| 19 | X | 26,498,601 | X | 26,580,441 | INV | Somatic | Splice variant |
| 22 | X | 26,730,476 | 5 | 12,878,258 | BND | Somatic | Transcript ablation |
| 23 | X | 27,138,447 | X | 27,875,254 | DEL | Somatic | Exon loss |
| 24 | X | 26,995,226 | X | 27,465,775 | DEL | Somatic | Exon loss |
| 7 | X | 27,634,320 | 3 | 53,899,511 | BND | Somatic | Bidirectional gene fusion |
| 19 | X | 26,528,430 | 23 | 27,291,280 | BND | Somatic | Bidirectional gene fusion |
| 19 | X | 26,529,617 | 23 | 27,291,403 | BND | Somatic | Gene fusion |
| 19 | X | 26,579,255 | 23 | 27,331,035 | BND | Somatic | Gene fusion and frameshift |
| 22 | X | 27,329,323 | 5 | 12,550,990 | BND | Somatic | Gene fusion and frameshift |

*SNV single nucleotide variant, SV structural variant*

confirm that, unlike human OS, translocations in *TP53* in canine OS are rare. Li−Fraumeni syndrome is not formally recognized as a cancer predisposition syndrome in the canine population Concordant with this observation, we identified one dog with a germline *TP53* point mutation[14]. While *RB1* mutations are frequently identified in human OS, *RB1* copy number loss is generally considered more prevalent in canine OS[4–6,14,16,43,44]. We found *RB1* copy number losses in 29%, although these were rarely homozygous events. We also found germline *RB1* variants in 58% of WGS and 46% of WES samples. Three WGS germline *RB1* variants had a concurrent *RB1* single copy number loss. Lastly, a recent study evaluating canine OS via WES reported 67% of Rottweilers carry potentially pathogenic germline *CDKN2A/B* variants[14]. Importantly, the similar incidence of *CDKN2B* germline variants observed in the present study suggest that this effect is not breed-restricted.

Signaling through the PI3K/AKT and MAPK pathways are hyper-activated in both canine and human OS[7,45]. We found somatic alterations in genes comprising the PI3K/AKT pathway in 37% of primary tumor samples, higher than the reported incidence in human OS[4]. Notably, inhibition of this pathway is associated with altered cell survival of both human and canine OS cell lines in vitro, supporting clinical efforts to target PI3K/AKT signaling in OS[4,46]. Similarly, MAPK pathway alterations were found in 17% of our canine OS samples. While these have not been associated with outcome in canine OS, MAPK pathway alterations have been identified as possible prognostic markers and therapeutic targets in human OS[47].

Assessment of somatic mutation signatures can provide clues to cancer etiology. In this study, the most common mutation signature corresponded to COSMIC 1 signature, which is prevalent in most human cancers[31,48,49]. Similar to previously published data, two mutation signatures described the variability within our WGS dataset[14]. Differences in the mutation signatures between the WGS and WES samples likely reflect a greater sequencing depth in the WES dataset. While the COSMIC signatures in our dataset do not directly overlap with those reported in human OS, there are shared characteristics between the signatures across both species. For example, BRCA signatures reported in human OS are known to generate characteristic patterns of kataegis[5]. It is therefore intriguing that the COSMIC 9 signature implicates AID processing cytidine deamination, which may provide similar support for genomic instability and kataegis in canine OS. Similarly, when comparing the WGS and WES datasets, fewer somatic mutations were found in the WES samples. Consistent with this, WGS has previously been shown to be a more powerful tool for identifying somatic coding alterations when compared to WES[50]. Genomic aberrations in key cancer-associated genes, including the PI3K/AKT pathway, MAPK pathway and epigenetic regulators, were identified across multiple large breed dogs. While the power of interrogating genomic aberrations in distinct dog breeds provides a unique opportunity to assess complex germline and somatic variants, the present study leveraged multiple dog breeds, creating a population with overall shorter linkage disequilibrium, reminiscent of that found in humans[51].

In both human and canine OS, there has been limited evaluation of the genomic landscape across matched primary and metastatic lesions. We found both shared and private mutations in a set of ten matched primary/metastatic canine OS samples. Somatic *TP53* mutations were the most commonly shared likely pathogenic drivers present in both primary and matched metastatic tumors. However, the majority of metastatic tumors demonstrated an increased somatic point mutation burden relative to their matched primary tumors (6/10) with 4/10 showing acquisition in the metastasis of candidate driver mutations not present in the matched primary (such as likely pathogenic *TP53* or *RB1* mutations), consistent with a model of branching clonal evolution and intra-patient heterogeneity.

RNA-seq of primary OS tumor samples was used to explore potential immune signatures. Specifically, we identified low expression of genes associated with innate immunity, complement activation, caspase-mediated apoptosis and T-cell activation. Consistent with these findings, a recent comparative transcriptome analysis in human, murine and canine OS found that decreased immune cell infiltration was associated with metastasis and poor survival in human OS, and immune cell changes were conserved across species[52]. Moreover, we observed significant differential expression of PD-L2 in the primary canine OS tumor samples which segregated with the major clades. Interaction of PD-1 on T cells with its ligands (PD-L1/PD-L2) typically expressed on antigen-presenting cells and tumor cells is an immune checkpoint and major driver of immune tolerance to tumor growth. Evaluation of pediatric solid tumors (including OS) for PD-1, PD-L1 and PD-L2 by immunohistochemistry has demonstrated low expression of these proteins[53]. Additionally, expression of PD-L1 as assessed by both mRNA and immunohistochemistry positively correlates with the presence of tumor-infiltrating lymphocytes (TILs) in human OS[54,55]. In keeping with this paradigm, we found decreased expression of genes associated with T-cell activation and chemotaxis that may reflect low numbers of TILs in OS samples, concordant with the low expression of PD-L2.

Dysregulation of SETD2, the sole histone methyltransferase catalyzing trimethylation of H3K36, has recently been implicated as a driver in both canine and human OS[14,56,57]. However, the functional significance of SETD2 inactivation is best described in renal cell carcinoma and leukemias in people where it has a tumor suppressor function[14,23,58–60]. It mediates numerous molecular processes involving gene regulation and the DNA damage response[23]. As *SETD2* mutations have only recently been identified in canine OS, a detailed characterization of how *SETD2* loss impacts the biology of OS has not yet been undertaken[14,56]. In their recent manuscript, Sayles et al. describe OS as a copy-number-driven cancer, and did identify one copy loss of SETD2, supporting the notion that SETD2 aberrations may be relevant in both species[36]. SETD2 can interact with and regulate p53, and we observed that *SETD2* and *p53* mutations largely co-occurred in our canine OS samples, with 8/10 samples with *SETD2* mutations having concurrent *TP53* mutations[61]. Additionally, when epigenetic mutations were evaluated in combination with *SETD2*, 67% of WGS samples demonstrated aberrations in epigenetic and chromatin-modifying genes. In support of this, somatic variations in KMT2C, a member of the ASC-2/NCOA6 complex (ASCOM) which possesses histone methylation activity, were found in 7/8 of human patients with high-grade OS, underscoring the role of epigenetic modulation in this disease[62].

The X-linked *DMD* gene encodes dystrophin, commonly associated with Duchene and Becker muscular dystrophy in both people and dogs. Recently, DMD has emerged as a potential tumor suppressor in several cancers, where deletions were associated with enhanced tumor cell migration, invasion and anchorage-independent growth[29]. Although inactivating DMD mutations have not been identified in human OS, WES of human OS samples described somatic *DMD* variants in 5/8 of patients. However, chromosomal deletions and rearrangements were not analyzed, and the pathogenicity of these variants was not investigated[63]. Striking similarities exist between *DMD* aberrations noted in our canine OS population and those reported in human mesenchymal tumors of myogenic origin. For example, somatic *DMD* deletions are found in both males and females, with common intragenic heterozygous mutations also noted[29]. Similar

intragenic *DMD* deletions were also recently described in human nonmyogenic sarcomas, supporting the notion that the *DMD* aberrations identified in canine OS are similarly relevant[64]. Prior studies showed that the active X chromosome is targeted in human sarcomas with heterozygous or homozygous *DMD* mutations, suggesting that even heterozygous somatic *DMD* deletions in canine OS may result in complete gene inactivation[29]. Lastly, consistent with known hotspots for *DMD* deletions in humans, all *DMD* SVs in our canine OS dataset occurred within the first 63 exons[29,65]. This is notable, as the Dp71 isoform is encoded in exons 63–79 and is ubiquitous in all cell types[66]. In *DMD*-deleted human sarcomas, the Dp71 isoform is maintained, while the 427-kDa isoform is lost in high-risk tumors and is embedded in the FRAXC common fragile site[29,67]. Additional work to determine the functional consequences of *DMD* loss in canine OS is ongoing.

In summary, our data confirm that the genomic complexity of canine OS resembles that of human OS. Conserved recurrent pathway aberrations are present that mirror many of the salient molecular features found in the human disease, providing further support for using dogs as a spontaneous large animal model of OS for therapeutic interrogation. Additionally, novel features of canine OS merit further exploration including the potential roles of *SETD2* and *DMD* in sarcoma initiation and progression.

## Methods

**Sample acquisition and library construction.** Fresh frozen tissues were collected via routine biobanking procedures at the time of surgery or humane euthanasia. Primary OS samples, required to be appendicular, were collected prior to definitive therapy (surgery, chemotherapy, radiation therapy). Samples for WGS were included only if dogs subsequently underwent surgery and chemotherapy and had follow-up annotation. Samples for WES were preferentially included if a matched primary appendicular and metastatic lesion were available. Sample collection occurred under the supervision of the attending veterinarian following institutional approvals (IACUC#: 2010A0015 (OSU), 16-6532A (CSU)) and informed consent from the pet owner. Tumor samples were confirmed to be OS and tumor content estimated by routine histologic evaluation of samples collected adjacent sequenced samples. DNA was isolated from normal muscle, whole blood, and primary OS tumor samples using the Qiagen DNeasy Blood & Tissue Kit. Total RNA was isolated from tumors and cell lines using the RNeasy Plus Mini Kit (both from Qiagen Inc., Hilden, Germany).

WGS, WES and RNA sequencing were performed on Illumina platforms with sample tracking by automated LIMS. For WGS ($n = 24$ primary OS samples), 150 ng of genomic DNA in 50 μL underwent fragmentation by acoustic shearing using a Covaris focused-ultrasonicator. Additional size selection was performed using a SPRI cleanup. WGS library preparation was performed using KAPA Biosystems' KAPA Library Prep Kit with Amplification Primer Mix and with palindromic forked adapters with unique 8 base index sequences embedded within the adapter (from IDT). Libraries were amplified by PCR and quantified using qPCR with probes specific to adapter ends. Libraries were normalized to 1.7 nM. Samples were pooled, underwent qPCR followed by combination with HiSeqX Cluster Amp Mix 1, 2 and 3 using the Hamilton Starlet Liquid Handling system. Cluster amplification of the templates was performed according to the manufacturer's protocol. Flowcells were sequenced on HiSeqX Sequencing-by-Synthesis Kits utilizing 151-bp paired-end reads, then analyzed using RTA2. WGS samples were sequenced to a mean coverage of 48× normal and 95× tumor (Supplemental Table 1). For WES ($n = 13$ primary OS, $n = 10$ metastatic OS), a custom Agilent SureSelect XT v1.6 canine exome capture kit with 982,789 probes covering 19,459 genes was used. Exome libraries were sequenced on the Illumina HiSeq4000 utilizing 82-bp paired-end reads. Output from Illumina software was processed by Picard to yield BAM files containing well-calibrated, aligned reads. WES samples were sequenced to a mean coverage of 140× normal and 146× tumor (Supplemental Table 1). RNA-seq library construction was performed using the TruSeq Strand Specific Large Insert Library protocol ($n = 24$ samples; sequenced at a depth of 152 million paired-end 101-base pair strand-specific reads per sample), TruSeq stranded total RNA library kit ($n = 18$; sequenced at a depth of 40 million paired-end 50-base pair strand-specific reads per sample) or TruSeq unstranded mRNA library kit ($n = 12$ samples; sequenced at a depth of 152 million reads per pair). RNA sequencing was performed using the Illumina protocol on Illumina HiSeqX and HiSeq4000 sequencers.

**Analysis of OS tumor and matched normal DNA.** Matched tumor and normal samples were processed through the workflow in Fig. 1. Tools, versions,

parameters, and references are provided in Supplemental Table 2. FastQs underwent quality control prior to alignment to the canine genome (CanFam3.1). Aligned BAMs were recalibrated with insertion−deletion realignment and duplicate marking[68]. Recurrently mutated genes were prioritized as having a likely role in canine osteosarcoma in this dataset based on incidence of the aberration. We additionally prioritized a list of genes known to be commonly mutated in human and canine osteosarcoma as described in Supplemental Table 4. Somatic and germline SNVs were identified by MuTect, Seurat and Strelka and those called by two or more callers were considered for final analysis[69–71]. Variants were annotated using SnpEff4.3T[72]. The SomaticSignatures R package was used to identify somatic mutation signatures in their trinucleotide context[73]. The KaryoploteR package was used to generate rainfall plots to identify areas of kataegis[74]. Germline mutations underwent additional filtering, excluding known common or benign single nucleotide polymorphisms as annotated in dbSNP and the Dog SNP database (DogSD). Annotation of variant impact was performed using Variant Effect Predictor (VEP)[75]. Structural variants were called from WGS data by DELLY v0.7 (https://github.com/dellytools/delly) and the somatic regions that passed quality control with an MAPQ score ≥40, Paired End ≥10 and Split Read ≥10 were analyzed. Structural variants identified by DELLY were classified as large chromosomal rearrangements (TRANS), inversions (INV), deletions (DEL) and duplications (DUP). CNVs were additionally detected in WGS data utilizing the tCoNuT algorithm (https://github.com/tgen/tCoNuT). The threshold for detection of copy number gains or losses was a $Log_2$ fold change of ≥0.4 to detect one copy gain and ≤−0.9 to detect two copy losses. Significant recurrent regions of copy number alteration were determined from tCoNuT results with the GISTIC 2.0 algorithm with a cut off score of G > 1.0 and a significance of Q < 0.05[76]. GraphPad Prism 7.04 (Graphpad Software, San Diego CA) was used for plots and statistics. Circos plots were created using the circos tool[77]. Additional tool parameters are referenced in Supplemental Table 2.

**RNA sequencing.** The 24 samples analyzed via WGS and 8 samples analyzed via WES were also subjected to RNA-seq. An additional 22 primary OS samples were included for a total of 54 primary OS samples analyzed via RNA-seq. RNA derived from a canine osteoblast cell line (CnOb; Cell Applications) was used as a comparator. FastQ files were aligned to CanFam3.1 using STAR align 2.4 to generate expression values (HTSeq counts) and determine relative transcript abundance (https://github.com/alexdobin/STAR). Additionally, these 24 primary OS tumor samples were compared to CnOb with DESeq2 using negative binomial generalized linear models[78]. Differentially expressed genes were filtered based on a $Log_2$ Fold Change ≤−2 or ≥2 and a Benjamini−Hochberg $p$-adjusted value of ≤1E-3. After filtering, HTSeq counts from these 302 statistically significant differentially expressed genes were utilized for hierarchical clustering across all samples with Spearman's rank correlation and pairwise average-linkage in GenePattern using gene- and sample-level normalization[79]. PANTHER pathway analysis was utilized to identify pathways associated within gene set aberrations (Supplemental Table 2)[80]. Two major clusters were observed in the hierarchical clustering heatmap. To test for significant differential gene expression between the two clusters, DESeq2 analysis (Supplemental Table 2) was conducted. Genes filtered by a Log2 Fold Change ≤−2 or ≥2 and a Benjamini−Hochberg $p$-adjusted ≤1E-5 were used to determine pathways in the two clades with PANTHER and DAVID (Supplemental Table 2)[80,81]. We additionally assessed correlation between gain-of-function or loss-of-function mutations in genes impacted by somatic structural mutations (CNVs and SVs) and gene expression values (TPMs). The average per-gene expression was calculated across samples as a threshold to determine relatively high or low expression among these cases and statistical significance was assessed by two-tailed $t$ tests.

**Statistics and reproducibility.** Data analysis and depiction of results was performed using GraphPad Prism v7.04 and RStudio (R3.5.0). A Mann−Whitney $U$ test was used to compare the mutation burden of primary and matched metastatic OS tumor samples. The threshold for detection of copy number gains or losses was a $Log_2$ fold change of ≥0.4 to detect one copy gain and ≤−0.9 to detect two copy losses. Significantly recurrent regions of copy number alteration were determined from tCoNuT results with the GISTIC 2.0 algorithm with a cut off score of G > 1.0 and a significance of Q < 0.05. Differentially expressed genes were filtered based on a $Log_2$ Fold Change ≤−2 or ≥2 and a Benjamini−Hochberg $p$-adjusted value of ≤1E-3. Genes filtered by a Log2 Fold Change ≤−2 or ≥2 and a Benjamini−Hochberg $p$-adjusted ≤1E-5 were used to determine pathways in the two clades with PANTHER and DAVID.

**Reporting summary.** Further information on research design is available in the Nature Research Reporting Summary linked to this article.

## Data availability

WGS, WES, and RNA-seq BAMs were deposited in the NCBI Sequence Read Archive (SRA accession: PRJNA525883). Additionally, all data used for figures are provided in the supplementary data items available as Excel spreadsheets referenced within this manuscript.

## Code availability

No custom code or algorithms were used in this study. All code and tools utilized are available through open source repositories or commercial (GISTIC) tools.

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

## Acknowledgements

Financial support for this research provided administrative supplements to the Dana-Farber/Harvard Cancer Center Support Grant (P30 CA006516) and the OSU Comprehensive Cancer Center Support Grant (P30 CA016058) from the National Cancer Institute. Additional support was provided by the UL1TR002733 from the National Center for Advancing Translational Sciences and by the Office of The Director and National Institutes of Health under Award Numbers P01CA165995-01A1 and K01OD019923. The content is solely the responsibility of the authors and does not necessarily represent the official views of the National Institutes of Health. We also thank Drs. William Kisseberth, Holly Borghese, Amy LeBlanc, and Christina Mazcko for assistance with sample collection, and Shukmei Wong and Alex Follette with assistance in data analysis.

## Author contributions

H.L.G. collected canine samples, assisted with data processing/analysis and interpretation, and wrote the majority of the manuscript. C.A.L. and K.A.J. were Project Directors for the administrative supplement to the DFCI/HCC, participated in the project conception and data interpretation and oversaw manuscript editing. W.P.D.H. participated in project conception and oversaw data analysis, interpretation, and manuscript preparation. J.M.T. and P.S. assisted with project conception. G.L. participated in project conception, provided samples for WES and RNA-seq analysis, and assisted with manuscript editing. J.F. provided RNA-seq data from additional canine primary OS samples. N.R. confirmed histopathologic diagnosis and evaluated histologic sections to determine tumor content. J.J. assisted with WGS and RNA-seq handling and assisted with manuscript preparation. P.H. assisted with data interpretation. W.L. oversaw WES and RNA-seq handling and processing. K.S., N.P., K.D., S.F., V.Z. and N.B. performed data analysis and interpretation and assisted with manuscript editing. R.R. and J.A. assisted with data analysis. S.L. provided additional samples for WES.

## Additional information

**Competing interests:** K.A.J. has received research funding from Amgen and Pfizer, and travel support from Loxo. C.A.L. has performed consulting work for The One Health Company, Anivive, Karyopharm therapeutics, Zoetis, and Blue Buffalo. W.P.D.H. has performed consulting work for The One Health Company, received research funding from Ethos Veterinary Health, and received travel support from Pathway Vet Alliance. The other authors declare no competing interests.

