## [Peer Review File · Communications Biology]

Reviewers' comments:

Reviewer #1 (Remarks to the Author):

Gardner et. al. present their work studying the genomic landscape (n = 37) and gene expression profiles (n = 54) of spontaneous canine osteosarcoma. They studied healthy tissues, primary tumor and metastatic samples and then compared their results with those of human osteosarcomas. Finally, by comparing the expression profiles of the tumors against those of a canine osteoblast cell line, they also found gene expression clusters of samples that revealed changes in immune-related genes. Overall the paper is well-written and the significance and potential impact of their work is well-explained. That being said, I have some comments.

Major

1 - The methods section is missing a lot of references: MuTect, Seurat, Strelka, SnpEff, SomaticSignatures, KaryoploteR, DELLY, Circos, DeSeq2, GenePattern...

2 - Where does the list of genes that the authors study come from? Is it a list of osteosarcoma human drivers? As of now it seems as if they chose to study some genes that, from the literature, they know could be related to osteosarcoma, but it's not really clear. If that is the case it would help to state it explicitly and give a table with the entire list.

3 - Related to the list of genes, I'd suggest doing an analysis with tools to detect cancer driver genes in their cohort, such as MutSigCV, MuSiC, OncodriveFML, e-Driver, 20/20+ etc. it is likely that some tools are human-specific, but if some can be applied to analyze this dataset I think it would be a great help. The sample size is not that big, but it might be enough to detect some strong signals (like TP53) and it would help to rationalize the list of genes presented here.

4 - Do the mutations detected in canine samples happen in equivalent positions as known oncogenic human mutations?

5 - The authors detect many CNVs and structural variants, do they find matching changes in gene expression levels in the RNAseq data? It'd help to make the case that these are indeed driver alterations.

6 - Does the breed of the dogs correlate with specific driver events?

Minor

1 - It is my opinion that the paper would really benefit from a first Figure showing the overall design of the experiment: which data and samples are available for which dog. As of now, we have the overall numbers for each category but it's difficult to follow what has been done with which samples. For instance, we know that 24 of the 54 samples with RNAseq data are from the WGS, but do the WES samples also overlap with the remaining 30?

2 - Does the mutation rate of the different genes correlate with that observed in humans?

3 - Does the presence of the germline variants in CDKN2A/B lower the age of incidence of the tumor? Does it correlate with changes in gene expression?

4 - I believe that most of the discussion of SETD2 actually belongs to the results section. Also, the authors talk about co-occurrence of SETD2 and TP3 alterations, but do not present any statistic.

Reviewer #2 (Remarks to the Author):

What are the major claims of the paper?

This manuscript has performed WGS, WES, and RNA-seq on a large number of canine OS tumors and metastases. The authors have reconfirmed a number of previous findings of sequencing efforts of canine OS, albeit in a more diverse set of breeds, and are the first to publish WGS of canine OS.

Are they novel and will they be of interest to the field?

On the surface it does not appear that there are many novel findings, though they did perform WGS on canine OS, which does not appear to be previously published. However, it nicely reconfirms findings of other sequencing efforts that focused on specific breeds, but instead here they have looked at a larger spectrum of breeds and found similar findings.

Is the work convincing?

Yes, the work is convincing and the results are not overstated, many observational statements.

Will the paper influence the thinking in the field?

As there is not a large number of novel findings in the paper, more so reconfirming previous findings, it is unlikely to have a large influence on the thinking in the field.

Appropriateness and validity of statistical analysis?

The statistical analyses appear appropriate and valid.

Reproducibility?

The methods used to analyze the genomics data and the pipeline of data processing are nicely outlined and will likely allow others to easily reproduce the findings from the data sets once they are made publically available.

Minor:

-“Additionally, we found chained rearrangements and complex chromosomal rearrangements involving multiple chromosomes that correlated with CNVs, suggestive of chromothripsis.”

In how many samples?

-“A cluster of thirty-one genes involved in aspects of the immune response was shown to have lower expression among most of the samples....”

Overall opinion:
Accept with minor revisions

Reviewer #3 (Remarks to the Author):

In this manuscript, Gardner et al. report the genetic landscape of canine osteosarcoma (OS). The canine OS is one of commonly used animal models of human OS to probe novel therapies as well as OS biology. This study has extended one previous canine OS study (Sakthikumar et al. *Cancer Res*, 2018 Jul 1;78(13):3421-3431) using a broad range of dog breeds, providing a much clearer genomic landscape, which fills current knowledge gaps. Through a comprehensive multi-omic (WGS, WES and RNA-seq) profiling of both primary and metastatic canine OS, authors demonstrate that canine OS recapitulates several genetic features of human OS including low point mutation burden (median 1.98 per Mb) and high translocation burden (median 9 per tumor). They found that consistent with previous reports in human OS, mutations in TP53, RB1, as well as multiple members of the PI3K/AKT and MAPK signaling pathways were identified in canine OS. They also found novel features of the canine OS genome including putatively pathogenic inactivating SETD2 (42%) and DMD (50%) aberrations. Overall, this is an important study revealing important new information regarding the genomic profile of canine OS, which provides a body of data for use in comparing genomic drivers between dogs and humans, and eventually facilitates therapeutic development. The manuscript is well prepared, except the following issues including:

1. In those 59 dogs, more than half of them are aged dogs (8-12 yr). It is similar to the prevalence to a second peak of age of human patients (60-80 yr). On other hand, in most of studies (WGS and WES) of human OSs, patients' tumor samples are pediatric osteosarcomas. It will be helpful if authors can compare two groups of dog samples (1-7 vs. 8-12 yrs) (e.g. the mutation rate)? Alternatively, they can discuss the differences (e.g. mutation rate or pattern) between those younger and older dog patients in the discussion section. For example, the lines 227-228 "Two samples analyzed via WES demonstrated a higher mutation rate in comparison to the remaining samples" have resulted from two dogs. Are they aged dogs?
2. The quality of the images in Fig.1 B & 1C is not good. Authors should provide better ones.
3. The data set should be available for reviewers (WGS, WES, and RNA-seq BAMs deposited in the NCBI Sequence Read Archive 194 (submission: 4844333)).
4. Authors stated, "the mutation burden in canine OS is relatively low", which is not accurate. Comparing to other pediatric cancers, the mutation burden of human and canine OS is relatively high (see Rickel et al., *Bone*. 2017;102:69-79). On the other hand, most of current human OS multi-omics studies have applied pediatric OS tumors whereas, in this study, there are more than 50% samples coming from aged dogs. It will be helpful if authors discuss this observation that the mutation burden between human and canine OSs are similar.
5. If authors representatively pricked up the dogs to show in Fig. 1B-C, it may be helpful to show dog #14 and #18 (the highest and lowest Total number of Somatic Coding Structural Variants).
6. If authors representatively pricked up the dogs to show in Fig. 2D, it may be helpful to show dog #7 and #11 (the highest and lowest Total number of Somatic Coding Single Nucleotide Variants).
7. Lines 223-235 that stated "A subset of samples (4/24 WGS)...(Figure 2D)" is not correct. Authors should address this.
8. Mutation signatures COSMIC1B and COSMI9 were detected in WGS analysis. It may be helpful if authors compare (or discuss) those signatures with ones previously reported from human osteosarcomas.
9. In a recent study for copy number variants using human OS tumors from Sweet-Cordero group (*Cancer Discov*. 2019 Jan;9(1):46-63.), copy number losses of SETD2 were identified, but DLG2 was not. It will be helpful if authors give an explanation for those differences and commonalities between

human and dog.

10. It is curious to know the status of gain of Notch signaling pathway in canine OSs given recent reports that (1), NOTCH3 (11% by patient) is one of the most commonly amplified genes together with MYC (39%) and CCNE1 (33%) (Sayles et al., *Cancer Discov.* 2019 Jan;9(1):46-63); (2), Notch hyperactivation may serve as a driver for murine osteosarcomagenesis (Tao et al., *Cancer Cell.* 2014 Sep 8;26(3):390-401).

11. For Supplemental Table 10, it will be helpful to present Excel file instead of PDF file.

12. Line 322, the statement "In contrast, both point mutations and structural variants in TP53 are frequently noted in human OS." needs reference (s).

Reviewed by Jianning Tao at Sanford Research

**Reviewer #1:
Major Comments**

1 - The methods section is missing a lot of references: MuTect, Seurat, Strelka, SnpEff, SomaticSignatures, KaryoploteR, DELLY, Circos, DeSeq2, GenePattern...

All references for the computational tools used in this study are now provided in Supplemental Table 2. Due to the limited number of references permitted, we elected to distribute references in this manner.

2 - Where does the list of genes that the authors study come from? Is it a list of osteosarcoma human drivers? As of now it seems as if they chose to study some genes that, from the literature, they know could be related to osteosarcoma, but it's not really clear. If that is the case it would help to state it explicitly and give a table with the entire list.

In keeping with this important recommendation, we have clarified our approach within the manuscript results and methods. We first sought to identify the most commonly mutated genes agnostic to the known role of the gene in dog or human osteosarcoma – for example, both SETD2 and DMD mutations were originally prioritized based solely on the prevalence and pattern of their mutation within the dataset. We then prioritized a list of genes known to be commonly mutated in human and canine osteosarcoma as described in the results section. We have also now included a supplemental table (Table S3) with the gene list used in this study.

3 - Related to the list of genes, I'd suggest doing an analysis with tools to detect cancer driver genes in their cohort, such as MutSigCV, MuSiC, OncodriveFML, e-Driver, 20/20+ etc. it is likely that some tools are human-specific, but if some can be applied to analyze this dataset I think it would be a great help. The sample size is not that big, but it might be enough to detect some strong signals (like TP53) and it would help to rationalize the list of genes presented here.

We agree that use of cohort-level statistical tools to detect significantly recurrently mutated drivers is important for agnostic and systematic identification of putative cancer genes that are recurrently mutated in large cohorts. In keeping with this recommendation, we have modified and implemented GISTIC to detect significant recurrent regions of copy number alteration (see Methods and Results). For SNVs, we have explored implementation of MutSigCV, MuSiC, OncodriveFML, e-Driver, and others and have found that most of these tools listed will only operate with the human reference genome without significant modifications to the underlying code. Although MuSiC has been reported to be operable in canine cancer genomics data, the older publicly available version is not amenable for analysis as 22 of the key dependencies needed to adapt MuSiC to canine are not currently available. We continue to work to modify these tools for implementation in canine data. However, in the interim, we believe that for detection of candidate canine osteosarcoma genes, the combined view of recurrently mutated genes plus osteosarcoma-associated genes described in the response to Comment #3 in combination with the addition of GISTIC-derived important CNV regions provides a comprehensive view of these alterations in this cohort, particularly given the cohort size.

4 - Do the mutations detected in canine samples happen in equivalent positions as known oncogenic human mutations?

We have now provided additional information on equivalent positions in human orthologs based on protein alignments for the key mutated genes we describe. We have added this detail in our discussion section and for select genes in Supplemental Table 6. For tumor suppressors, truncating point mutations tend to occur in various locations across the gene, but with some shared damaging hotspots as is also observed in human cancers. For oncogenes, we do observe human-equivalent hotspots in our data.

5 - The authors detect many CNVs and structural variants, do they find matching changes in gene expression levels in the RNAseq data? It'd help to make the case that these are indeed driver alterations.

In order to address this important point, we have assessed correlation between expression levels and presence/absence of putative driver mutations in 12 of the recurrently mutated cancer genes we highlight in Figure 1. These data are now described in the results. We found significant correlations between TPM values and MYC gains as well as between TPM values and SETD2 loss. Importantly, other events not measured here may also affect dysregulation of some of these important cancer genes (e.g. epigenetic marks) and drive over- or under-expression such as that seen in a number of cases with relatively low levels of CDKN2A, but no identified inactivating mutation.

6 - Does the breed of the dogs correlate with specific driver events?

We did not detect correlations between breed and driver mutations, although our cohort contained a heterogeneous collection of breeds and thus only very large effects would have been captured in this study if they had existed. We now make this point in our results section.

Minor Comments

1 - It is my opinion that the paper would really benefit from a first Figure showing the overall design of the experiment: which data and samples are available for which dog. As of now, we have the overall numbers for each category but it's difficult to follow what has been done with which samples. For instance, we know that 24 of the 54 samples with RNAseq data are from the WGS, but do the WES samples also overlap with the remaining 30?

We have clarified this information by including the overlapping RNA-seq samples in the oncoprint in Figure 1. All samples not listed in Figure 1 were used for RNA-seq only.

2 - Does the mutation rate of the different genes correlate with that observed in humans?

Yes, for many of the mutated genes, the incidence and type of mutations are similar to those found in human osteosarcoma. The similarities and differences in type and incidence of mutations between canines and humans are described in the discussion.

3 - Does the presence of the germline variants in CDKN2A/B lower the age of incidence of the tumor? Does it correlate with changes in gene expression?

Germline CDKN2A/B variants were observed in both young and old dogs. Therefore, germline CDKN2A/B variants do not appear related to the age of incidence of OS. We did identify downregulation of CDKN2A/B expression by RNA-seq, suggesting that the germline deletions are damaging alterations, and have described this feature in the results section.

4 - I believe that most of the discussion of SETD2 actually belongs to the results section. Also, the authors talk about co-occurrence of SETD2 and TP3 alterations, but do not present any statistic.

SETD2 and TP53 alterations co-occurred in 8/10 (80%) tumors with SETD2 mutations. We have updated the manuscript to include this information. However, we do not have the statistical power ($p=0.15$; Fisher's exact test) to make more than a descriptive statement. We have reorganized elements of the results section to more succinctly describe the landscape of key recurrent drivers (e.g. SETD2, TP53, and DMD) as well as more clearly subdivide the results describing the primary versus metastatic sample analysis. We hope that this will help address the concerns around description of SETD2 data.

Reviewer #2:

Minor Comments:

-“Additionally, we found chained rearrangements and complex chromosomal rearrangements involving multiple chromosomes that correlated with CNVs, suggestive of chromothripsis.” In how many samples?

These changes were identified in 9/24 samples. This information is included in the manuscript text. We additionally now provide circos plots for all WGS samples as an additional supplemental figure (Figure S5) in order to facilitate visualization of these events

-“A cluster of thirty-one genes involved in aspects of the immune response was shown to be have lower expression among most of the samples....”

The authors are unsure what question Reviewer #2 is asking. We are happy to address any question regarding the RNAseq data should the reviewer provide clarification. We additionally direct the reviewer to new quantitative analysis of the major clades identified by unsupervised clustering in addition to pathway analysis of genes significantly differentially expressed between these clades.

Reviewer #3:

1. In those 59 dogs, more than half of them are aged dogs (8-12 yr). It is similar to the prevalence to a second peak of age of human patients (60-80 yr). On other hand, in most of studies (WGS and WES) of human OSs, patients' tumor samples are pediatric osteosarcomas. It will be helpful if authors can compare two groups of dog samples (1-7 vs. 8-12 yrs) (e.g. the mutation rate)? Alternatively, they can discuss the differences (e.g. mutation rate or pattern) between those younger and older dog patients in the discussion section. For example, the lines 227-228 “Two samples analyzed via WES demonstrated a higher mutation rate in comparison to the remaining samples” have resulted from two dogs. Are they aged dogs?

We appreciate the reviewers' comments on the role of age in our dataset. In dogs, osteosarcoma very commonly develops around 7-10 years of age. As indicated in the manuscript, most dogs in our study fall within this age range, with a small subset developing the disease at a young age (1-2 years). We highlighted this feature because it shows that our dataset is representative of what is expected clinically in spontaneous osteosarcoma in dogs. We acknowledge that dogs tend to develop the disease at a relatively older age. However, the

genomic landscape, response to treatment, and development of treatment-resistant metastases otherwise closely recapitulate the human condition. Thus, we do not feel that this discrepancy in age of onset is, in itself, a defining feature of the disease. Nonetheless, we have evaluated our entire dataset based on age-bracket (as shown in Fig 1) and are unable to make definitive conclusions regarding mutation rate and age. This is due to the fact that only one dog in the WGS and one dog in the WES dataset were from the 1-4yr age bracket, and in both cases the mutation rate/pattern and incidence/proportion of SVs (for WGS dog) were close to the median for the whole population. In response to this important comment by this reviewer, we have now included a more comprehensive discussion of age, but feel that the disproportionately small number of dogs in the dataset younger than 4yr would be difficult to justify. Regarding the two WES samples with an increased incidence of missense SNVs, these two dogs were in the higher age bracket, and we have indicated this in the manuscript text.

2. The quality of the images in Fig.1 B & 1C is not good. Authors should provide better ones.

The circos plots have been replaced with higher quality images.

3. The data set should be available for reviewers (WGS, WES, and RNA-seq BAMs deposited in the NCBI Sequence Read Archive 194 (submission: 4844333)).

The dataset has been made available to the reviewers through SRA. The BAMs associated with this work can be accessed via the following link:

[ftp://ftp-](ftp://ftp-trace.ncbi.nlm.nih.gov/sra/review/SRP187781_20190312_123858_679f47e946645e35bc0d606f07a1217e)

[trace.ncbi.nlm.nih.gov/sra/review/SRP187781_20190312_123858_679f47e946645e35bc0d606f07a1217e](ftp://ftp-trace.ncbi.nlm.nih.gov/sra/review/SRP187781_20190312_123858_679f47e946645e35bc0d606f07a1217e)

4. Authors stated, “the mutation burden in canine OS is relatively low”, which is not accurate. Comparing to other pediatric cancers, the mutation burden of human and canine OS is relatively high (see Rickel et al., Bone. 2017;102:69-79). On the other hand, most of current human OS multi-omics studies have applied pediatric OS tumors whereas, in this study, there are more than 50% samples coming from aged dogs. It will be helpful if authors discuss this observation that the mutation burden between human and canine OSs are similar.

We thank the reviewer for these comments. We have added further language to add clarity based on the context of pediatric tumors versus adult tumors. We agree that the mutation burden identified in our study population is similar to that reported in human OS, and is relatively high in the context of other pediatric tumors but relatively low in the broader context of adult cancers.

5. If authors representatively pricked up the dogs to show in Fig. 1B-C, it may be helpful to show dog #14 and #18 (the highest and lowest Total number of Somatic Coding Structural Variants).

The circos plots in Fig 1 were selected as representative cases with and without TP53 mutation. However, we have taken the reviewers suggestion and replaced them with #14 and 18 in the primary figure. We also agree that it is helpful to show a representative spectrum of SV burden and have thus now included all circos plots for WGS samples as Supplementary Figure 5.

6. If authors representatively pricked up the dogs to show in Fig. 2D, it may be helpful to

show dog #7 and #11 (the highest and lowest Total number of Somatic Coding Single Nucleotide Variants).

The two rainfall plots shown were chosen to represent evidence of kataegis in our dataset. However, we agree that more broadly visualizing high- and low-SNV samples in this cohort would be helpful and thus have now included a supplementary figure (S2) containing all rainfall plots for the WGS samples.

7. Lines 223-235 that stated “A subset of samples (4/24 WGS)...(Figure 2D)” is not correct. Authors should address this.

We have clarified these lines with “A subset of samples (4/24 WGS), two of which are shown in Figure 2D...” There were four samples with evidence of kataegis in our dataset. We elected to show two of the rainfall plots in Figure 2D.

8. Mutation signatures COSMIC1B and COSMI9 were detected in WGS analysis. It may be helpful if authors compare (or discuss) those signatures with ones previously reported from human osteosarcomas.

The COSMIC1 signature is present in most human cancers, including OS. We have included a discussion of the BRCA signatures reported in human OS (similar to signatures 3 and 5). While there is little direct signature overlap, there were shared characteristics between our canine dataset and human OS signatures. Namely, the BRCA signatures reported in human OS are known to generate characteristic patterns of kataegis. Interestingly, the COSMIC 9 signature implicates AID processing cytidine deamination, and may provide further support for genomic instability and kataegis in canine OS. We have discussed these connections in more detail in the discussion.

9. In a recent study for copy number variants using human OS tumors from Sweet-Cordero group (Cancer Discov. 2019 Jan;9(1):46-63.), copy number losses of SETD2 were identified, but DLG2 was not. It will be helpful if authors give an explanation for those differences and commonalities between human and dog.

We appreciate this critical insight and have added these elements to our discussion. In their recent manuscript, Sayles et al. described OS as a copy-number driven cancer. While SETD2 was present in the supplementary copy number tables of this publication (almost always only one-copy loss), SETD2 was not mentioned within the manuscript itself. Comparative copy number loss of DLG2 in both canine (56%) and human (42%) OS was recently described in a separate manuscript (Shao et al. Oncogene 2018).

10. It is curious to know the status of gain of Notch signaling pathway in canine OSs given recent reports that (1), NOTCH3 (11% by patient) is one of the most commonly amplified genes together with MYC (39%) and CCNE1 (33%) (Sayles et al., Cancer Discov. 2019 Jan;9(1):46-63); (2), Notch hyperactivation may serve as a driver for murine osteosarcomagenesis (Tao et al., Cancer Cell. 2014 Sep 8;26(3):390-401).

Mutations in Notch were extremely rare within our dataset (1 missense variant in a metastatic sample; 1 translocation in Notch2 a primary tumor sample). Only 4 samples had copy number gains in Notch (Notch 1, Notch 2, Notch4). This data is available in the supplementary tables. Given the low incidence and diversity of Notch isoforms with copy number gains, they were not

described as a separate entity within the manuscript, as we did not feel there was sufficient justification to discuss all events occurring at a low incidence within our dataset.

11. For Supplemental Table 10, it will be helpful to present Excel file instead of PDF file.

When files were uploaded for manuscript submission, all excel files were required to be converted to PDF format. We are happy to provide excel files for all supplementary tables through the editor.

12. Line 322, the statement “In contrast, both point mutations and structural variants in TP53 are frequently noted in human OS.” needs reference (s).

The requested reference has been added to the manuscript.

We hope that these changes address all concerns raised regarding this manuscript.

REVIEWERS' COMMENTS:

Reviewer #1 (Remarks to the Author):

The authors have successfully addressed most of my questions. I think that the comparison of the mutation patterns between dogs and humans is now much more clear as is the rationale for the genes that they focused on.

My only minor comment is that I still think that the methods used in the data analysis should be referenced in the main manuscript. The choice of the tools used is fundamental to the results of the paper and the developers of these tools should receive proper credit for them.

Reviewer #2 (Remarks to the Author):

All of my concerns have been addressed and I think the responses to the other reviewer critiques were also addressed.

Reviewer #3 (Remarks to the Author):

The authors have addressed the concerns. This revision has been strengthened with adding data and changes. The reservations on the previous revision have been removed.